Behavioral niche partitioning in a sympatric tiger beetle assemblage and implications for the endangered Salt Creek tiger beetle

Brosius Tierney R. 1 TierneyBrosius@augustana.edu
Higley Leon G. 2
1 Department of Biology, Augustana College , Rock Island, IL , USA
2 Biochemistry Hall, University of Nebraska-Lincoln , Lincoln, NE , USA
Somers Michael
Electronic publication date: 2013 Sep 17
Publication date: 2013
Volume: 1
Electronic Location ID: e169
Received 2013 Jul 23; Accepted 2013 Aug 31
Copyright: © 2013 Brosius et al.
Copyright year: 2013
Copyright holder: Brosius et al.
License: This is an open access article distributed under the terms of the Creative Commons Attribution License, which permits unrestricted use, distribution, and reproduction in any medium, provided the original author and source are credited.
License URL: https://creativecommons.org/licenses/by/3.0/

Keywords: Cicindelidae, Thermoregulation, Ecophysiology, Competitive exclusion, Endangered species, Conservation biology

Funding: Nebraska Game and Parks Commission US Fish and Wildlife Service This research was supported in part by grants from the Nebraska Game and Parks Commission and the US Fish and Wildlife Service. The funders had no role in study design, data collection and analysis, decision to publish, or preparation of the manuscript.

==============================
How behavioral patterns are related to niche partitioning is an important question in understanding how closely related species within ecological communities function. Behavioral niche partitioning associated with thermoregulation is well documented in tiger beetles as a group. Co-occurring species of salt flat tiger beetles have adapted many thermoregulatory behaviors to cope with this harsh ecosystem. On first examination these beetles appear to occur in overlapping microhabitats and therefore compete for resources. To determine if behavioral niche partitioning is allowing multiple species to occur within the same harsh salt flat ecosystem we observed Cicindela nevadica lincolniana, Cicindela circumpicta, Cicindela fulgida, and Cicindela togata between 8:00 h and 21:00 h and recorded all behaviors related to thermoregulation using a digital voice recorder. Results of this study strongly indicate that competition among these species for resources has been reduced by the adaptation of different thermoregulatory behaviors such as spending time in shallow water, avoiding the sun during the hottest parts of the day, and by positioning their body against or away from the soil. The endangered C. n. lincolniana appears to rely most heavily on the shallow water of seeps for their diurnal foraging behavior (potentially limiting their foraging habitat), but with the advantage of allowing foraging during the hottest times of the day when potential competitors are less frequent. Ironically, this association also may help explain C. n. lincolniana’s susceptibility to extinction: beyond the loss of saline wetlands generally, limited seeps and pools even within remaining saline habitat may represent a further habitat limitation within an already limited habitat.

Introduction

Competition is one of the main forces driving niche partitioning and, consequently, speciation. Interspecific competition may lead to segregation among or within habitats followed by adaptations to different microhabitats (Schultz & Hadley, 1987). Ultimately, adaptation through selection is genetic but the phenotypic expression of selection may take many forms. Although we commonly think of physiological and morphological changes as products of selection through niche partitioning, variation in the behavior of sympatric species also can be a mechanism for niche partitioning.

Harsh environments, where the biota is reduced and physiological adaptation is essential for survival, present ideal laboratories for examining the interplay of adaptation and competition. In such systems, the strong selection pressure associated with the physical environment provides a stage upon which interspecific competition plays. Saline wetlands, and their associated salt flats are one such harsh environment.

The saline wetlands of eastern Nebraska are home to a unique cast of adapted organisms. Along with saline requirements in these organisms’ life histories, many also are adapted to tolerate the harsh, desert-like environment typically associated with salt flats during the summer months (Farrar & Gersib, 1991). Within this environment exists a large assemblage of congeneric, sympatric tiger beetle species, including the endangered Cicindela nevadica lincolniana (Willis, 1967; Spomer & Higley, 1993; Spomer, Higley & Hoback, 1997). Because of the uniqueness of this environment and the endangered status of one of these beetle species much attention has been given to this group of tiger beetles.

Behaviors that serve to partition resources and reduce physiological stress have been examined for many species of tiger beetles, Cicindelidae, but much remains to be discovered about the diversity and functions of tiger beetle behavior (Pearson & Mury, 1979; Pearson, 1980; Pearson & Stemberger, 1980; Pearson & Knisley, 1985; Ganeshaiah & Belavadi, 1986; Schultz & Hadley, 1987; Hoback et al., 2000; Hoback, Higley & Stanley, 2001; Romey & Knisley, 2002). Past and current research supports the theory that these tiger beetles are using oviposition as a mechanism for niche partitioning (Hoback et al., 2000; Hoback, Higley & Stanley, 2001; Allgeier, 2005). Tiger beetle larvae do not move more than a few cm from where their eggs are originally deposited by the female. Among three salt marsh tiger beetle species (C. n. lincolniana, C. circumpicta, and C. togata) we observed oviposition differences based on soil salinity (Allgeier, 2005; Brosius, 2010). This unique life history trait, along with the larval dependence on limited prey resources for proper development (Mury Meyer, 1987), emphasizes the importance of the location chosen for oviposition.

Prey also are essential to adult female fecundity. The amount of prey consumed by adult females is directly tied to their ability to lay greater numbers of eggs (Pearson & Knisley, 1985). The ability to lay more eggs can be important to the success of individual populations given high mortality in the larval stages. For example, Shelford (1913) documented the mortality of some populations of Cicindela scutellaris to be as high as 80% due to the parasitoid Anthrax sp., and Knisley & Schultz (1997) documented a mortality rate upwards around 63% from tiphiid wasps.

Because females must gain enough caloric resources for egg production and development, it is likely that adult tiger beetle assemblages have evolved behaviors to reduce interspecific competition as adults. In particular, beyond direct competition for prey (e.g., Hoback, Higley & Stanley, 2001), competition may be mediated through partitioning of foraging habitats: spatially, seasonally, or temporally. In this partitioning, temperature tolerance to the foraging habitat may play a key role.

There is a long history of thermoregulation studies that focus on tiger beetles and that link thermoregulation behaviors to resource partitioning (Dreisig, 1980; Dreisig, 1981; Dreisig, 1984; Dreisig, 1985; Morgan, 1985; Pearson & Lederhouse, 1987; Schultz & Hadley, 1987; Schultz, Quinlan & Hadley, 1991; Schultz, 1998; Hoback, Higley & Stanley, 2001; Romey & Knisley, 2002). Many tiger beetles are found in environments where temperatures are capable of exceeding 60°C, a lethal temperature for most insect species (Hadley, 1994). From these studies it is clear that Cicindela are capable of regulating their body temperatures by changing their body orientation and shuttling between microclimates. Adult tiger beetles spend a high percentage of their day balancing foraging behavior with thermoregulatory behavior. Pearson & Stemberger (1980) determined that the gain of one hour of additional foraging could increase the biomass of ingested prey by as much as 20%. This increase in prey would translate into an increase in egg production for adult females.

Physiological character divergence in species’ ability to cope with temperatures could be a mechanism to reduce intraguild predation. Differences in heat tolerances between species is a likely mechanism to reduce competition, however, a close examination of lethal maximum temperature of 13 tiger beetle species near Willcox, Arizona, USA revealed very few differences between upper heat tolerances among species (Pearson & Lederhouse, 1987). Out of the 13 species examined in this study only two were found to be significantly different and the difference was less than one degree from the overall mean of 48.1°C. Hoback, Higley & Stanley (2001) determined the lethal maximum temperature for two species of tiger beetles (Cicindela circumpicta and Cicindela togata) within the complex of co-occurring species in the eastern saline wetlands of Nebraska and found no differences between species, with results similar to values observed by Pearson & Lederhouse (1987) with different Cicindela species. While tiger beetle species may have some variation in their ability to cope with different degrees of humidity and temperature (Pearson & Lederhouse, 1987; Schultz & Hadley, 1987; Schultz, Quinlan & Hadley, 1991) physiological differences cannot account for all behavioral differences observed in the field.

To determine the evolutionary cause for such differences, Hoback, Higley & Stanley (2001) investigated lethal high temperatures, prey base, prey size, mobility, and the effect of direct predation of C. togata by C. circumpicta. Laboratory studies that investigated feeding behavior of C. togata in both the absence and presence (separated by a clear pane of glass) of C. circumpicta indicated that C. togata feeding behavior was negatively affected by the presence of C. circumpicta. These results coupled with field observations strongly indicated that intra-guild predation was possible among salt marsh tiger beetles.

In the case of C. n. lincolniana and its co-occurring Cicindela species it is likely that similar evolutionary forces are at work as seen in earlier studies are at work. Past research strongly suggests that behaviors associated with the reduction of predation, thermoregulation, foraging, and predator avoidance may reduce competition among sympatric, adult tiger beetles (Pearson & Mury, 1979; Schultz & Hadley, 1987; Pearson & Juliano, 1991; Romey & Knisley, 2002). Consequently, we examined diurnal behavior of foraging salt marsh Cicindela species, with particular focus on potential differences in thermoregulatory behaviors. In particular, given the endangered species status of C. n. lincolniana, we were interested in identifying any unique adaptations that might contribute to population declines different from the other salt marsh tiger beetle species.

Methods and Materials

During the summer of 2007 initial observations were made of three species of saline adapted tiger beetles: C. circumpicta, C. togata, and, C. n. lincolniana at the Arbor Lake Complex, Lincoln, NE. From these initial observations it appeared that these tiger beetles exhibit a large variety of behaviors and use a wide range of microhabitats. Behaviors associated with thermoregulation were of particular interest. Based on the 2007 data, an ethogram or catalogue of discrete behaviors was developed. Observations were classified specifically as states, events, and behaviors.

“States” described physical aspects of the environments, here, temperature, light, and substrate. Temperature included measures of soil-surface temperature, 1 cm above the soil (comparable to tiger beetle body height), and ambient (1 m) air temperatures. Light indicated if the subject was standing in the sun or shade. Substrate indicated if the subject was on dry soil, mud, or shallow water (typically on water near seeps or on algal mats along stream margins).

“Behaviors” included running, stilting, basking, and standing. Stilting occurs when the tiger beetle holds itself up off of the substrate with its front two legs extended straight downward. Often the beetles appear to be standing at a 45° angle from the vertical. Stilting occurs during the hottest time of day in an effort to reduce surface area and keep bodies away from the hot soil surface (Pearson & Vogler, 2001). Basking occurs when the tiger beetle presses its body up to the substrate in an effort to warm its body in the early morning hours (Pearson & Vogler, 2001).

“Events” were recorded when individuals exhibited a behavior that had no measurable time duration. Events included mandible dipping (dipping mandibles into the substrate beetles were are standing on), wing pumping (a quick opening and closing of their elytra), flight (this event had a measurable time duration but we almost always lost track of the individual), and abdomen dipping (the individual would dip its abdomen into the water by doing what appeared to be a quick pushup).

Behaviors of four co-occurring species of tiger beetles were examined on 20 June 2008, 30 June 2008, 7 July 2008, 23 June 2009, and 2 July 2009. Dates were chosen based on the predicted weather. We chose days that were predicted to have average or above average temperature with no precipitation. Species included C. circumpicta and C. togata, C. n. lincolniana, and the spring-fall species C. fulgida (which occasionally occurred in early summer). Behavior, states, and events were recorded using digital voice recorders and later transcribed using the program JWatcher™ (Version 1.0). One recording was made for each hour in the field.

For each hour, three or four observers were randomly assigned a species to observe. Recordings were made from the first individual of that species the observer could locate using close-focus binoculars. At the start of each recording the observer noted the date, time, species, and, observer name. The observer watched one individual as long as they could in a 30 min period. If recordings were less than 10 min in length the observer recorded a second individual of the same species for that hour. On the rare occasion that no species of that individual could be found (after at least 15 min of looking) the observer selected the first tiger beetle that they could find for recording observations. Along with these behaviors hourly temperatures were recorded. The surface temperature was taken from the area near where the recordings were being made. Temperature measurements were taken at 1 m and 1 cm elevation from the same location. Because tiger beetle’s bodies are approximately 1 cm above the soil surface, we used 1 cm measurement to reflect the temperatures being experienced by the beetles.

The data were entered into the program JWatcher. For each block of time, we averaged time spent exhibiting individual behaviors (basking, running, standing, and stilting), time spent in sun or shade, and time spent on type of substrate (mud, dry soil, and water). Because times that each individual was watched varied, we converted the time spent to percentages for species comparisons. Events were averaged by hour by species.

Analysis

Statistical differences were examined by using non-parametric approaches, because data were not normally distributed and could not be transformed to meet normality requirements. We used non-parametric procedures in GraphPad InStat (version 3.10 for Windows, GraphPad Software, San Diego California USA, www.graphpad.com) for analyses. These procedures included Mann-Whitney tests or non-parametric ANOVA with Kruskal-Wallis Test (corrected for ties) for overall significance, and Dunn’s Multiple Comparison for separating species where the Kurskal-Wallis indicated a significant overall effect. The logic of statistical choices and limitations follow that of Motulsky (2009).

Responses to state and behavior variables were analyzed by determining the mean time in a state for two periods: morning (8:00–11:00) and afternoon (11:00–21:00). The decision to subdivide the day into these two periods was made a priori based on the expectation of substantial temperature and behavioral differences at these times of day (based on published research and our own previous observations). Events were analyzed based on mean counts per day (i.e., the number of occurrences of each event). States, behaviors, and events were analyzed across species and within species. For analysis purposes the combination of day and different individuals within a species represents our replications. Additionally, randomly assigning different observers to record behaviors for different species each hour was used to minimize any potential bias among observers recording behaviors.

Results

States

State variables of association with temperature, light, and substrate differed with time of day and among species (Figs. 1–4). Temperatures increased until 15:00 in all measurements (Fig. 1). As a result of the absorption of solar radiation, soil temperature became higher than ambient and 1 cm air temperatures after 11:00. At 08:00 the surface temperature was more than 1°C cooler than 1 cm above the surface and ambient temperature. By 15:00 the surface temperature was 7.9°C higher than at 1 cm above the surface. At 08:00 there was almost no difference between ambient temperature and 1 cm above the soil surface (0.12°C). At 16:00 h the difference rose to 1.9°C. Temperatures dropped dramatically between 20:00 and 21:00. At 1 cm above the salt flat surface, temperatures dropped 5.4°C.

Figure 1 Average salt flat temperatures by hour.

Temperatures were recorded on the salt flats where observations were made for adult tiger beetles. Temperatures were recorded one meter above the soil surface to determine ambient temperature, one cm above the soil surface to determine the air temperature the subjects were experiencing, and at the soil surface.

Figure 2 Time spent in sun or shade by tiger beetle species.

Proportion of time spent in the sun or the shade between 8:00 and 21:00 h (bars on graph) and soil surface temperature of the salt flats (dotted line) over those hours for four species of salt flat tiger beetle: (A) C. circumpicta, (B) C. n. lincolniana, (C) C. fulgida, and (D) C. togata.

Figure 3 Time spent on different substrates by tiger beetle species.

Proportion of time spent on wet soil (mud), dry soil, and in shallow water between 8:00 and 21:00 h (bars on graph) and soil surface temperature of the salt flats (dotted line) over those hours for four species of salt flat tiger beetle: (A) C. circumpicta, (A) C. n. lincolniana, (C) C. fulgida, and (D) C. togata.

Figure 4 Time spent engaging in thermoregulatory behavior.

Proportion of time spent in four distinct behaviors related to thermoregulation between 8:00 and 21:00 h (bars on graph), and soil surface temperature of the salt flats (dotted line) over those hours for four species of salt flat tiger beetle: (A) C. circumpicta, (B) C. n. lincolniana, (C) C. fulgida, and (D) C. togata.

Time spent in the sun was directly linked to the time of day and, therefore, temperature. For all species time spent in the sun was significantly higher in the morning (08:00–11:00) when the salt flat temperatures were the coolest (Fig. 2, Table 1). As temperatures rose the amount of time spent in the shade increased for all species (Fig. 2). Significant differences in percentages of time spent in the sun were found between species. Cicindela nevadica lincolniana spent the most time in the sun throughout the day (77.1% of their total time) (Fig. 2, Table 1).

Table 1 Comparison of proportion of time spent in the sun by C. circumpicta, C. fulgida, C. n. lincolniana, and C. togata.

Morning and afternoon/evening differences across species were evaluated with a Mann-Whitney test. Species differences (within morning or afternoon/evening) were determined by Kruskal-Wallis Test through non-parametric ANOVA (K-W test statistic corrected for ties and P determined from chi-square distribution). Significant species comparisons were determined by Dunn’s Multiple Comparisons Test (only results with P < 0.05 are shown).

Proportion of Time in Sun	
8:00–21:00	M-W U	P > F	
Comparison: 8:00–11:00 vs. 11:00–21:00	719.5	<0.0001	
Times	n	Mean	sd			
8:00–11:00	28	0.92	0.14			
11:00–21:00	139	0.69	0.29			
8:00–11:00	K-W (df = 3)	P > X2	
Comparison: species × species	1.43	ns	
Species	n	Mean	sd			
C. circumpicta	7	0.91	0.21			
C. fulgida	7	0.94	0.10			
C. n. lincolniana	11	0.90	0.14			
C. togata	3	0.98	0.03			
11:00–21:00	K-W (df = 3)	P > X2	
Comparison: species × species	9.07	= 0.0283	
C. fulgida vs. C. n. lincolniana		<0.05	
Species	n	Mean	sd			
C. circumpicta	41	0.67	0.23			
C. fulgida	34	0.54	0.32			
C. n. lincolniana	50	0.74	0.26			
C. togata	14	0.62	0.41			

Cicindela togata spent the most amount of time on dry soils (Fig. 3, Table 2). Cicindela nevadica lincolniana spent significantly more time standing on the damp surfaces and in the shallow water of the seeps than the other species observed (Fig. 3, Table 2). Both C. circumpicta and C. fulgida spent the majority of their time on damp substrate as opposed to dry soil (Figs. 3A and 3C, Table 2). Cicindela circumpicta spent more time on dry substrate when temperatures rose in the late afternoon (Fig. 3A, Table 2). Cicindela fulgida spent the most amount of their time on damp substrate (60.1% total) and time spent on damp substrate increased with temperature (Fig. 3C, Table 3).

Table 2 Comparison of proportion of time spent on dry surfaces by C. circumpicta, C. fulgida, C. n. lincolniana, and C. togata.

Morning and afternoon/evening differences across species were evaluated with a Mann-Whitney test. P values for Mann-Whitney are estimated (not exact) because of tied ranks. Species differences (within morning or afternoon/evening) were determined by Kruskal-Wallis Test through non-parametric ANOVA (K-W test statistic corrected for ties and P determined from chi-square distribution). Significant species comparisons were determined by Dunn’s Multiple Comparisons Test (only results with P < 0.05 are shown).

Proportion of Time on Dry Substrate	
8:00–21:00	M-W U	P > F	
Comparison: 8:00–11:00 vs. 11:00–21:00	1514.0	= 0.0255	
Times	n	Mean	sd			
8:00–11:00	28	0.14	0.33			
11:00–21:00	139	0.69	0.29			
8:00–11:00	K-W (df = 3)	P > X2	
Comparison: species × species	0.7793	ns	
Species	n	Mean	sd			
C. circumpicta	7	0.14	0.37			
C. fulgida	7	0.14	0.37			
C. n. lincolniana	11	0.09	0.30			
C. togata	3	0.33	0.57			
11:00–21:00	K-W (df = 3)	P > X2	
Comparison: species × species	42.042	<0.0001	
C. circumpicta vs. C. n. lincolniana		<0.001	
C. circumpicta vs. C. togata		<0.05	
C. fulgida vs. C. n. lincolniana		<0.01	
C. fulgida vs. C. togata		<0.01	
C. n. lincolniana vs. C. togata		<0.001	
Species	n	Mean	sd			
C. circumpicta	41	0.46	0.48			
C. fulgida	34	0.38	0.49			
C. n. lincolniana	50	0.04	0.20			
C. togata	14	0.62	0.34			

Table 3 Comparison of proportion of time spent on mud by C. circumpicta, C. fulgida, C. n. lincolniana, and C. togata.

Morning and afternoon/evening differences across species were evaluated with a Mann-Whitney test. P values for Mann-Whitney are estimated (not exact) because of tied ranks. Species differences (within morning or afternoon/evening) were determined by Kruskal-Wallis Test through non-parametric ANOVA (K-W test statistic corrected for ties and P determined from chi-square distribution). Significant species comparisons were determined by Dunn’s Multiple Comparisons Test (only results with P < 0.05 are shown).

Proportion of Time on Mud	
8:00–21:00	M-W U	P > F	
Comparison: 8:00–11:00 vs. 11:00–21:00	1645.5	ns	
Times	n	Mean	sd			
8:00–11:00	28	0.70	0.41			
11:00–21:00	139	0.58	0.45			
8:00–11:00	K-W (df = 3)	P > X2	
Comparison: species × species	5.900	ns	
Species	n	Mean	sd			
C. circumpicta	7	0.85	0.38			
C. fulgida	7	0.81	0.38			
C. n. lincolniana	11	0.86	0.42			
C. togata	3	0.67	0.58			
11:00–21:00	K-W (df = 3)	P > X2	
Comparison: species × species	16.717	= 0.0008	
C. circumpicta vs. C. togata		<0.05	
C. fulgida vs. C. togata		<0.01	
C. n. lincolniana vs. C. togata		<0.001	
Species	n	Mean	sd			
C. circumpicta	41	0.54	0.48			
C. fulgida	34	0.61	0.49			
C.n. lincolniana	50	0.72	0.35			
C. togata	14	0.13	0.34			

Another important difference in habitat use was indicated by associations with standing in water (at margins of seeps and streams). Of the species observed, during the hottest parts of the day C. n. lincolniana spent over a third of the time in water, while other species spent little or no time in water (Fig. 3, Table 4).

Table 4 Comparison of proportion of time spent on water by C. circumpicta, C. fulgida, C. n. lincolniana, and C. togata.

Morning and afternoon/evening differences across species were evaluated with a Mann-Whitney test. P values for Mann-Whitney are estimated (not exact) because of tied ranks. Species differences (within morning or afternoon/evening) were determined by Kruskal-Wallis Test through non-parametric ANOVA (K-W test statistic corrected for ties and P determined from chi-square distribution). Significant species comparisons were determined by Dunn’s Multiple Comparisons Test (only results with P < 0.05 are shown).

Proportion of Time on Water	
8:00–21:00	M-W U	P > F	
Comparison: 8:00–11:00 vs. 11:00–21:00	1703.0	ns	
Times	n	Mean	sd			
8:00–11:00	28	0.15	0.30			
11:00–21:00	139	0.08	0.22			
8:00–11:00	K-W (df = 3)	P > X2	
Comparison: species × species	13.274	= 0.0041	
C. circumpicta vs. C. n. lincolniana		<0.05	
Species	n	Mean	sd			
C. circumpicta	7	0	0			
C. fulgida	7	0.05	0.12			
C. n. lincolniana	11	0.36	0.40			
C. togata	3	0	0			
11:00–21:00	K-W (df = 3)	P > X2	
Comparison: species × species	44.754	<0.0001	
C. n. lincolniana vs. C. circumpicta		<0.001	
C. n. lincolniana vs. C. fulgida		<0.001	
C. n. lincolniana		<0.001	
Species	n	Mean	sd			
C. circumpicta	41	0	0			
C. fulgida	34	0	0.03			
C. n. lincolniana	50	0.23	0.32			
C. togata	14	0	0			

Behaviors

All four species of tiger beetle spent a large portion of their time alternating between standing and running (Fig. 4). This reflects typical tiger beetle foraging behavior where they run in short bursts or stand watching for prey (Pearson & Juliano, 1991). Cicindela togata had the largest shift between running and standing in the middle of the day (Fig. 4D). Basking was almost always associated with morning hours (before 11:00) and evening hours (after 16:00), with more than three times more basking in the morning than later. Across species, basking occurred 15.5% of the time before 11:00 versus 5.1% of time after 11:00 (Mann-Whitney U = 13.86, P < 0.0001). This difference lined up with the coolest hours of the day. Stilting was almost completely associated with the middle parts of the day for all four species. Cicindela circumpicta spent more time basking in the early hours (Fig. 4A).

Events

The total events per hour were averaged over the entire day for all four species (Table 5). Cicindela nevadica lincolniana was the only species to display abdomen dipping into water (Table 5). Similarly, C. n. lincolniana averaged 78.05 mandible dips per hour, which was far greater than C. circumpicta which was the next highest at 5.44 mandible dips per hour. Cicindela togata averaged 0.88 flights per hour, which was greater than C. fulgida which was the next highest at 0.51 flights per hour. Cicindela togata appeared to make many short flights during the 8:00 time block but short flights seemed to have no other significant correlation to time of day for any other tiger beetle species (Fig. 5). Wing pumping occurred in greater frequency in C. n. lincolniana earlier in the day but appeared to occur with greater frequency later in the day for C. circumpicta (Table 5).

Figure 5 Average mandible dips, wing pumps, and flight events per hour.

(A) Average recorded mandible dipping events per hour of observation for C. n. lincolniana and C. circumpicta (bars on graph), and soil surface temperature of the salt flats over those hours (dotted line); (B) Average recorded wing pumping events per hour of observation for C. n. lincolniana, C. fulgida, and C. circumpicta (bars on graph), and soil surface temperature of the salt flats (dotted line) over those hours; and (C) Average recorded flight events per hour of observation for C. n. lincolniana, C. togata, and C. circumpicta and surface temperature of the salt flats over those hours.

Table 5 Means and standard deviations (sd) of total behavioral events per hour averaged daily (11:00–21:00 for C. circumpicta, C. fulgida, C. n. lincolniana, and C. togata.

Significant differences within an event were determined by Kruskal-Wallis Test through non-parametric ANOVA (K-W test statistic corrected for ties and P determined from chi-square distribution, df = 3). Significant species comparisons were determined by Dunn’s Multiple Comparisons Test (only results with P < 0.05 are shown).

		Abdomen Dipping	Mandible Dipping	Wing Pumping	Flying	
Species	n	mean	sd	mean	sd	mean	sd	mean	sd	
C. circumpicta	41	0	0	5.4	10.3	4.0	4.8	0.20	0.46	
C. fulgida	34	0	0	2.5	4.2	5.2	5.4	0.59	1.02	
C. n. lincolniana	50	0.24	1.2	75.8	99.6	3.9	4.4	0.10	0.30	
C. togata	14	0	0	5.8	9.7	3.9	3.5	0.57	1.16.5	
Comparison		K-W	P > X2	K-W	P > X2	K-W
w (df = 3)	P > X2	K-W
w (df = 3)	P > X2	
		—	—	55.50	<0.0001	0.626	ns	9.807	0.02	
C. n. lincolniana vs. C. circumpicta				<0.001					
C. n. lincolniana vs. C. fulgida				<0.001				<0.05	
C. n. lincolniana vs. C. togata				<0.01					

Discussion

The high surface temperatures reached by the salt flats in the middle of the day are a challenge for organisms living in this ecosystem, including tiger beetles. The results of this study suggest that these species have evolved multiple mechanisms for coping with the high temperatures found on the salt flats. Because these mechanisms vary among species, these results imply that co-occurring species of adult tiger beetles within this system are segregating their foraging through behavioral differences associated with temperature.

Shade-seeking behavior during the heat of the day is a common behavior seen in many organisms. Interestingly, C. n. lincolniana is very active during the hottest part of the day while other species of salt flat tiger beetles spend much of their time seeking refuge in the shade. This difference is probably not due to differences in physiology, but rather associated with differences in their behavior. Unlike other species, C. n. lincolniana spends over a third of its time foraging in shallow seeps. During this foraging, what we observed as mandible dipping is probably two different types of behavior. Tiger beetles drink water by sinking their mandibles into a damp substrate. We observed tiger beetles taking a few seconds to hydrate using this behavior, however, when mandible dipping C. n. lincolniana would frequently come up with a small insect larva that it would quickly consume. Even in the hottest part of the day C. n. lincolniana was able to forage while in the shade of the salt grass growing in the seeps. Additionally, the seemingly unique behavior of abdomen dipping by C. n. lincolniana suggests that wetting the abdomen might be another mechanism for heat reduction.

Differences in habitat use and foraging behavior allow C. n. lincolniana to be active during the hottest part of the day. This unique foraging behavior probably explains the large difference in mandible dipping frequency between species (Table 4, Fig. 5A). There also was a temporal component to foraging by C. n. lincolniana as they would move out of the water and further out onto the salt flats as evening approached. We observed few other tiger beetle species after 19:00 h.

Cicindela fulgida was the only species of tiger beetles we observed that is classified as a spring-fall species. Spring-fall species of tiger beetle emerge as adults in the fall and overwinter as adults in contrast to summer species which overwinter as larvae and emerge as adults in the spring. The adult beetles of this species that we were observing had overwintered as adult beetles and had been active as early as March (possibly even warm days in February). We expected that C. fulgida would have a more difficult time dealing with extreme temperatures, and C. fulgida did spend the most time on damp substrate and in the shade (Figs. 2C and 3C). This species also spent a large proportion of the morning hours basking (Fig. 4C), presumably in an effort to warm their body temperature.

Cicindela circumpicta spent more time on dry substrates as temperatures rose during the day. Most of the shade on the salt flats is provided by vegetation which grows on the perimeter of the flat. This area has dry substrate because of the distance from the seep. As the temperatures dropped later in the day C. circumpicta appeared to move back onto damp surfaces to forage (Fig. 3A). Cicindela circumpicta also spent a large proportion of their time basking in the early morning hours (Fig. 4A). It is possible that the larger body mass of C. circumpicta requires more basking to raise its body temperature.

Cicindela togata spent the most time on dry surfaces. Unlike C. circumpicta who moved to dry surfaces during the hottest part of the day and moved back to damp surfaces C. togata appears to spend all of its time on dry soils. The time spent on damp soils in the morning is likely a result of the lack of dry soil due to morning condensation. Cicindela circumpicta and C. togata have the same tolerance to heat yet they forage in different microhabitats on the flats (Hoback, Higley & Stanley, 2001). Hoback, Higley & Stanley (2001) theorized that microhabitat differences in foraging was a consequence of C. togata avoiding being preyed upon by C. circumpicta. In support of this notion, laboratory studies demonstrated that C. togata’s behavior is modified by the presence of C. circumpicta. Because C. circumpicta is the largest of the salt marsh tiger beetles, other species may have adapted different foraging behaviors to avoid contact with C. circumpicta to avoid predation as well as to reduce competition in foraging.

An alternative (or possibly complementary) explanation for C. togata use of microhabits was suggested by one of our anonymous reviewers. Because C. togata has the smallest body size and longer legs than other tiger beetles we studied, C. togata can heat and cool more rapidly than other species (Pearson & Vogler, 2001). Given this advantage, C. togata may be able to use dry substrates when foraging more effectively than other species (who may have great thermoregulation issues).

Wing pumping by tiger beetles is thought to be a thermoregulatory behavior to release heat (Pearson & Vogler, 2001), but C. togata (which had the highest per hour average) pumped its wings at a greater frequency in the morning. Cicindela fulgida and C. circumpicta both appeared to pump their wings with greater frequency in the middle of the day but did not show a direct association with temperature (Table 5).

Conclusions

How tiger beetles allocate their time in relation to temperature has been studied extensively (Dreisig, 1980; Dreisig, 1981; Dreisig, 1984; Dreisig, 1985; Morgan, 1985; Pearson & Lederhouse, 1987; Schultz & Hadley, 1987; Schultz, Quinlan & Hadley, 1991; Schultz, 1998; Hoback, Higley & Stanley, 2001; Romey & Knisley, 2002). Our results indicate that temperature, as well as potential competitive relationships among Cicindelida species, appears to be tied into what substrate beetles chose to occur on, whether or not they chose to spend time in the sunlight, and what behaviors they exhibited.

In contrast to what might be expected for the endangered member of this assemblage, C. n. lincolniana clearly demonstrates behaviors that afford it advantages in accommodating high temperatures and avoiding potential competition with other tiger beetles. Key to these behaviors is the species reliance on shallow water of the seeps for their diurnal foraging behavior (potentially limiting their foraging habitat), but with the advantage of allowing foraging during the hottest times of the day when potential competitors are less frequent. Indeed, by frequenting water, C. n. lincolniana was able to remain active in sunlight far more than other species, and maintain its activity over a longer portion of the day.

The endangered status of C. n. lincolniana requires that those involved with the conservation of this insect examine its habitat requirements closely. Because this beetle appears to have reduced competition over food resources by adapting to forage in a unique environment they may be more susceptible to habitat destruction. Organisms that are highly specialized, such as C. n. lincolniana, are thought to be more susceptible to extinction due to habitat destruction (Kammer, Baumiller & Ausich, 1997; Kotiaho et al., 2005). The shallow seeps found in saline wetlands have been destroyed by the channelization of these water ways over the last 100 years. Consequently, the close association of C. n. lincolniana with seeps and associated shallow pools seems to let C. n. lincolniana adults forage at times when temperatures may limit foraging for other saline-adapted tiger beetles. Ironically, this association also may help explain C. n. lincolniana’s susceptibility to extinction: beyond the loss of saline wetlands generally, limited seeps and pools even within remaining saline habitat may represent a further habitat limitation within an already limited habitat.

Supplemental Information

Table S1 Raw Data on salt marsh tiger beetle behavioral thermoregulation

Click here for additional data file.

We thank Drew Tyre for suggestions regarding data analysis, and Lauren Thompson, Mitch Paine, and Carmen Mostek for their assistance with recording behavior data (on some very hot days!). We also thank Lauren for her diligence in helping transcribe hours of behavior recordings. Finally, we appreciate the helpful comments of our anonymous reviewers, particularly the suggestions regarding C. togata foraging behaviors and microhabitat choice.

Additional Information and Declarations

Competing Interests

Author Contributions

The authors declare that they have no competing interests.

Tierney R. Brosius conceived and designed the experiments, performed the experiments, analyzed the data, wrote the paper.

Leon G. Higley conceived and designed the experiments, analyzed the data, contributed reagents/materials/analysis tools, wrote the paper.

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
