# Peer review of "Behavioral niche partitioning in a sympatric tiger beetle assemblage and implications for the endangered Salt Creek tiger beetle"

_PeerJ, doi:10.7717/peerj.169_

## Round 0.1 · original submission · Minor Revisions

I have read the MS and find it well written and almost ready for acceptance. Please consider the minor issues as given by the referees.

Reviewer 1 ·

Basic reporting

No comments

Experimental design

This manuscript by Higley and Brosius is well written and presents a well- designed study of a rare tiger beetle that has important implications for its conservation. The introduction presents the relevant literature and an effective background to this study. The data presented and the statistical analysis of these behaviors seems appropriate in providing valid evidence of difference among species.

Validity of the findings

The conclusion that these determined differences in behavior can reduce competition for food seems well supported and justified. The different behaviors of C. n. lincolniana provide a valid insight to its adaptation to the saline habitats where it is found, and can inform management by protecting habitat with seeps and other features it requires

Additional comments

I would offer an alternate or complementary explanation for the foraging microhabitat of togata. It is significantly smaller than the other species, with long legs and consequently can both extend higher above the substrate where it is slightly cooler and would have reduced heat loading and quicker cooling. In tiger beetles (and other insects) smaller body size can result is more effective behavioral thermoregulation that species with larger body size, thus allowing for foraging on warmer substrates or during the warmer parts of the day (Pearson and Vogler).

Reviewer 2 ·

Basic reporting

The authors introduce a very complex set of interactions in an understandable context of ecology and conservation. The English is solid with a minimum of unnecessary jargon. A simple but significant change should be made in lines 59 and 61. The endangered tiger beetle is not an endangered species. It is an endangered population or recognized subspecies. There are a few typos throughout (e.g., line 339 "on" should be inserted between reliance and shallow).

Experimental design

No comments

Validity of the findings

No comments

Additional comments

The authors successfully integrate a complex set of interactions and test them convincingly to accomplish both goals of ecology and conservation.

---

## Round 0.2 · accepted · Accept

Thank you for the rvision. The MSreads well.